# A Robust $\widetilde{\mathcal{O}}(1/\sqrt{T})$ Rate for TD Learning with Linear Function Approximation without Projections

## Abstract

We investigate the finite-time convergence properties of Temporal Difference (TD) learning with linear function approximation, a cornerstone algorithm in the field of reinforcement learning. We are interested in the so-called "robust" setting, where the convergence guarantee does not depend on the minimal curvature of the potential function. While prior work has established convergence guarantees in this setting, these results typically rely on the assumption that each iterate is projected onto a bounded set, a condition that is both artificial and does not match the current practice. In this paper, we challenge the necessity of such an assumption and present a refined analysis of TD learning. For the first time, we show that the simple projection-free variant converges with a rate of $\widetilde{\mathcal{O}}(\frac{\|\boldsymbol{\theta}^*\|_2^2}{\sqrt{T}})$, even in the presence of Markovian noise. Our analysis reveals a novel self-bounding property of the TD updates and exploits it to guarantee bounded iterates.

## 1 Introduction

Temporal Difference (TD) learning (Sutton, 1988) is a cornerstone of modern reinforcement learning. It provides a model-free approach to policy evaluation, estimating the value function of a given policy within a Markov Decision Process (MDP). The versatility of TD methods has led to applications in diverse domains, including games (Silver et al., 2016), robotics (Gu et al., 2017), and autonomous systems (Chen et al., 2015). At its core, TD learning iteratively updates value function estimates based on the difference between predictions at successive time steps.

Despite its conceptual simplicity and widespread use, the theoretical analysis of TD learning, particularly with linear function approximation in large state spaces, presents considerable challenges. Early seminal work by Tsitsiklis and Van Roy (1996) established asymptotic convergence conditions by framing TD as a stochastic approximation algorithm. More recently, understanding the non-asymptotic behavior and finite-time performance of TD has become an active area of research. Challenges primarily arise from the correlated nature of samples generated by the underlying Markov chain, which can introduce bias and dependencies into the learning updates.

Several studies have provided finite-time analyses under various assumptions on the potential function, on a projection step, and on the stepsize. In particular, two *complementary* kinds of analyses are known, giving rise to a "robust" convergence rate of $\widetilde{\mathcal{O}}(1/\sqrt{T})$[1] (e.g., Bhandari et al., 2018; Liu and Olshevsky, 2021) or to "fast" one of $\widetilde{\mathcal{O}}(1/T)$(e.g., Bhandari et al., 2018; Srikant and Ying, 2019; Patil et al., 2023; Samsonov et al., 2024; Mitra, 2024). These two rates are complementary because the hidden constant in the fast rate depends on the inverse square of the curvature of the potential function which, while always present, can be arbitrarily small. Instead, the $\widetilde{\mathcal{O}}(1/\sqrt{T})$ robust rate is independent of the curvature. Hence,

---

[1]$\widetilde{\mathcal{O}}$ hides polylogarithmic terms and may also hide dependencies on the mixing time.

Table 1: Summary of convergence rates and assumptions for linear TD learning. $\omega$ is the minimal curvature of the objective function, all other quantities are defined in Section 3.

| Rate | Paper | Algorithm Design | | Bound independent of $\lambda_{\min}(\Phi^\top D\Phi)$ |
|---|---|---|---|---|
| | | Hyperparameters | Without Projection | |
| $\widetilde{\mathcal{O}}(1/T)$ | Bhandari et al. (2018) | $\omega,\ \phi_\infty$ | ✗ | ✗ |
| | Srikant and Ying (2019) | $\omega,\ \tau(\eta),\ \phi_\infty,\ \|\boldsymbol{\theta}^*\|_2,\ T$ | ✓ | ✗ |
| | Patil et al. (2023) | $\tau(1/4),\ \phi_\infty,\ T$ | ✓ | ✗ |
| | Samsonov et al. (2024) | $\tau(1/4),\ \phi_\infty,\ T$ | ✓ | ✗ |
| | Mitra (2024) | $\omega,\ \tau(\eta),\ \phi_\infty,\ T$ | ✓ | ✗ |
| $\widetilde{\mathcal{O}}(1/\sqrt{T})$ | Bhandari et al. (2018) | $\tau,\ \phi_\infty,\ T$ | ✗ | ✓ |
| | Liu and Olshevsky (2021) | $\tau,\ \phi_\infty,\ T$ | ✗ | ✓ |
| | Sun et al. (2021) | $\phi_\infty$ | ✗ | ✗ |
| | **This paper**, Theorem 4.2 | $\alpha,\ \phi_\infty,\ T$ | ✓ | ✓ |

in non-asymptotic regimes, the fast rate can be arbitrarily worse than the robust one.[2] This mirrors what happens in the stochastic approximation setting, and it is well-known that in practice the robust rate can be preferable (Nemirovski et al., 2009).

In this work, we focus on the need for a projection step to achieve robust rates. In fact, for fast rates, the assumption of minimal curvature leads to a contraction that simplifies the analysis, eliminating the need for a projection. However, there are no known results on unprojected TD to achieve the robust $\widetilde{\mathcal{O}}(1/\sqrt{T})$ rate. Indeed, as far as we know, it was previously unknown whether this was even possible. This projection step, while common for simplifying the analysis (Kushner, 2010), can be limiting, as it modifies the algorithm used in practice and may not always be practical or desirable.

**Contributions.** In this paper, for the first time, we provide a finite-time analysis of TD(0) with linear function approximation under Markovian observations *without the requirement of iterate projection and proving a rate independent of the curvature of the potential function.* Our main contribution is to demonstrate that learning rates of the form $\frac{1}{\sqrt{t}\ln^2 T}$ are sufficient to guarantee a self-bounding property of TD: The iterates are constrained, in expectation, to a bounded domain around the optimal solution. Moreover, our analysis differs fundamentally from those that aimed to prove the update is a noisy contraction. We also show a convergence rate $\widetilde{\mathcal{O}}(\frac{\|\boldsymbol{\theta}^*\|_2^2}{\sqrt{T}})$ for the potential that guides the convergence of the TD algorithm, as defined in Liu and Olshevsky (2021). Table 1 summarizes[3] and compares our results with existing finite-time analyses of TD with linear function approximation.

## 2 RELATED WORK

The initial theoretical understanding of how TD learning with linear function approximation converges over time was established by Tsitsiklis and Van Roy (1996), who framed TD methods as stochastic approximation algorithms (Kushner, 2010). That work did not derive finite-time convergence rates. Subsequent research (Korda and La, 2015; Lakshminarayanan and Szepesvari, 2018; Dalal et al., 2018) did provide such rates, but a significant limitation was the assumption that data are drawn independently from the stationary distribution. In practice, data are typically collected sequentially along a single trajectory of the Markov chain, introducing temporal correlations between samples. These correlations make analyzing even the basic TD(0) method challenging.

Bhandari et al. (2018) offered the first finite-time analysis for TD learning under the more realistic Markovian data, drawing parallels to stochastic gradient descent. However, their analysis, as well as Liu and Olshevsky (2021), requires a projection step to control the magnitude of the iterates/updates. Sun et al. (2021) examined Adam-inspired (Kingma,

---

[2]See Appendix A for an in-depth discussion of the literature on this point.
[3]See Appendix B for a precise discussion of the hyperparameters for the algorithms

2014) adaptive TD variants, but they require a projection as well. Here, we remove the need to project, while obtaining the same $\widetilde{\mathcal{O}}(1/\sqrt{T})$ rate of Bhandari et al. (2018).

Another line of work started from taking advantage of the curvature of the potential function. This allowed Srikant and Ying (2019) to be the first to provide finite-time error bounds for TD learning with linear function approximation under Markovian sampling without a projection step, using a control-theoretic approach based on Lyapunov theory. While elegant, the analysis in Srikant and Ying (2019) relies on learning rates that depend on the strong-convexity (curvature) parameter of the potential function. Given that the strong convexity is typically unknown, this means that this result only implies the existence of a good but unknown learning rate. A closely related analysis with similar limitations was presented in Mitra (2024). Subsequently, Patil et al. (2023) removed the dependence of the stepsize on this strong-convexity parameter, yielding a more practical algorithm, but under the but with a price of a data-dropping variant of TD. Later, Samsonov et al. (2024) improved the analysis of Patil et al. (2023) to obtain high-probability bounds in Markovian settings. Sun et al. (2022) extended the fast analysis to neural networks in the NTK regime. However, in all these results, the non-asymptotic convergence rate can become arbitrarily slow from bad linear mappings. We discuss this caveat in more detail in Section 5. Our proof method is fundamentally different from the above ones, which removed the projection by proving a contraction. Instead, we show that the iterates are bounded for reasons analogous to what happens in Stochastic Gradient Descent (SGD). In fact, SGD can have bounded iterates even for non-strongly convex objectives, as shown, for example, by Xiao (2010); Orabona and Pál (2021); Ivgi et al. (2023) under various update schemes and assumptions on the potential and stepsizes.

Another minor difference with prior work is our choice of the potential function: We study the potential function proposed in Liu and Olshevsky (2021), which improves earlier formulations by adding a term proportional to the discount factor $\gamma$.

## 3 Notation and Assumptions

We briefly review the required background on Markov Decision Processes (MDPs) and on TD learning with linear function approximation. For a comprehensive treatment of these topics, the reader may consult Sutton et al. (1998) and Mannor et al. (2022).

### 3.1 Discounted Markov Decision Processes

A *discounted-reward MDP* is a tuple $(\mathcal{S}, \mathcal{A}, \boldsymbol{P}, r, \gamma)$ where $\mathcal{S} = \{s[1], s[2], \ldots, s[n]\}$ is a finite state space, $\mathcal{A}$ is a finite action space, $P(s' \mid s, a)$ denotes the transition probability from state $s$ to state $s'$ under action $a$, $r(s, a, s') \in \mathbb{R}_{\geq 0}$ is a deterministic reward, and $\gamma \in (0, 1)$ is the discount factor. We denote the start state by $s_0$ and the state at time $t > 0$ by $s_t$.

In this paper, we focus on the common task in reinforcement learning to evaluate a policy with respect to the expected discounted sum of rewards, $\boldsymbol{V}$.

**Policies and Value functions.** A *stationary policy* is a mapping $\mu : \mathcal{S} \to \Delta^{|\mathcal{A}|-1}$ and the value function $\boldsymbol{V}^\mu$ associated with $\mu$ is given by $V^\mu(s_0) = \mathbb{E}[\sum_{t=0}^{\infty} \gamma^t r(s_t, a_t, s_{t+1}) \mid a_t \sim \mu(s_t)]$, where $s_0 \in \mathcal{S}$.

**Induced Markov chain.** A policy $\mu$ induces the transition matrix

$$P^\mu(s, s') = \sum_{a \in \mathcal{A}} \mu(s, a) P(s' \mid s, a), \qquad s, s' \in \mathcal{S} .$$

Throughout, $\mu$-induced rewards are shortened to $r(s, s') := \sum_a \mu(s, a) r(s, a, s')$ and are assumed bounded by $r_\infty$. The Bellman operator associated with $\mu$, defined as

$$(T^\mu \boldsymbol{V})(s) := \sum_{s'=1}^{n} P^\mu(s, s') \left( r(s, s') + \gamma V(s') \right), \qquad \boldsymbol{V} \in \mathbb{R}^n,$$

is a $\gamma$-contraction on $(\mathbb{R}^n, \|\cdot\|_\infty)$; hence $\boldsymbol{V}^\mu$ is its unique fixed point.

## 3.2 Mixing and Matrix norms

To study the finite-time behavior of the Markov chain, we impose the following standard ergodic condition.

**Assumption 1.** *The Markov chain induced by policy $\mu$ with transition matrix $\boldsymbol{P}^\mu$ is irreducible and aperiodic.*

Under Assumption 1 the chain admits a unique stationary distribution $\boldsymbol{\pi} \in \Delta^{n-1}$ with $\boldsymbol{\pi}\boldsymbol{P}^\mu = \boldsymbol{\pi}$, and mixes geometrically:

**Theorem 3.1** (Levin and Peres 2017, Thm. 4.9)**.** *There exist constants $1 < C \le 2$ and $\alpha \in [1/2, 1)$ such that $\max_{s \in \mathcal{S}} \left\| (\boldsymbol{P}^\mu)^t(s, \cdot) - \boldsymbol{\pi} \right\|_{\mathrm{TV}} \le C\,\alpha^t$, for $t \ge 0$.*

We denote the mixing time $\tau(\epsilon)$ as $\min\{t \in \mathbb{N} \mid C\alpha^t \le \epsilon\}$ and $\tau := \tau(1/\sqrt{T})$.

## 3.3 TD(0) with Linear Function Approximation

Exact value computation is infeasible on large state spaces. So, we approximate $\boldsymbol{V}^\mu$ by a linear predictor and estimate its weight $\boldsymbol{\theta}$ with TD learning (Sutton, 1988).

**Linear architecture.** Let $\phi_i : \mathcal{S} \to \mathbb{R}$ for $i \in \{1, \dots, d\}$ be fixed feature mappings, we define the following vectors and matrices:

$$\boldsymbol{\phi}(s) := (\phi_1(s), \dots, \phi_d(s))^\top \in \mathbb{R}^d, \quad \boldsymbol{\Phi} := \begin{bmatrix} \boldsymbol{\phi}(s[1])^\top \\ \boldsymbol{\phi}(s[2])^\top \\ \vdots \\ \boldsymbol{\phi}(s[n])^\top \end{bmatrix} \in \mathbb{R}^{n \times d}, \quad V_{\boldsymbol{\theta}}(s) := \boldsymbol{\theta}^\top \boldsymbol{\phi}(s), \quad \boldsymbol{\theta} \in \mathbb{R}^d.$$

We recall the following standard assumption on the features (Bhandari et al., 2018; Srikant and Ying, 2019; Patil et al., 2023; Mitra, 2024; Samsonov et al., 2024).

**Assumption 2.** $\boldsymbol{\Phi}$ *has full column rank $d$, and $\|\boldsymbol{\phi}(s)\|_2 \le \phi_\infty$ for all $s \in \mathcal{S}$.*

**TD error and update.** Given the weight $\boldsymbol{\theta}_t \in \mathbb{R}^d$ and trajectory $(s_t, r(s_t, s_{t+1}), s_{t+1})$ at time $t$, the TD error is $\delta_t := r(s_t, s_{t+1}) + \gamma V_{\boldsymbol{\theta}_t}(s_{t+1}) - V_{\boldsymbol{\theta}_t}(s_t)$. TD(0) then performs

$$\boldsymbol{\theta}_{t+1} = \boldsymbol{\theta}_t + \eta_t\, \delta_t\, \nabla_{\boldsymbol{\theta}} V_{\boldsymbol{\theta}_t}(s_t) = \boldsymbol{\theta}_t + \eta_t \left( r(s_t, s_{t+1}) + \gamma \boldsymbol{\theta}_t^\top \boldsymbol{\phi}(s_{t+1}) - \boldsymbol{\theta}_t^\top \boldsymbol{\phi}(s_t) \right) \boldsymbol{\phi}(s_t) := \boldsymbol{\theta}_t + \eta_t \boldsymbol{g}_t\,.$$

with stepsize $\eta_t > 0$.

**Weighted norms and Dirichlet seminorm.** (Diaconis and Saloff-Coste, 1996; Ollivier, 2018; Liu and Olshevsky, 2021). Let $\boldsymbol{A} \succ 0$. For vectors $\boldsymbol{x}, \boldsymbol{y} \in \mathbb{R}^n$ define $\langle \boldsymbol{x}, \boldsymbol{y} \rangle_{\boldsymbol{A}} := \boldsymbol{x}^\top \boldsymbol{A} \boldsymbol{y}$ and $\|\boldsymbol{x}\|_{\boldsymbol{A}} := \sqrt{\langle \boldsymbol{x}, \boldsymbol{x} \rangle_{\boldsymbol{A}}}$. With $\boldsymbol{D} := \mathrm{diag}(\pi(1), \dots, \pi(n))$ we write

$$\langle \boldsymbol{V}, \boldsymbol{V}' \rangle_{\boldsymbol{D}} = \sum_s \pi(s)\, V(s)\, V'(s), \qquad \|\boldsymbol{V}\|_{\boldsymbol{D}}^2 = \sum_s \pi(s)\, V(s)^2\,.$$

The *Dirichlet seminorm* is defined as $\|\boldsymbol{V}\|_{\mathrm{Dir}}^2 := \frac{1}{2} \sum_{s,s'} \pi(s)\, P^\mu(s, s') \big( V(s') - V(s) \big)^2$.

**Projected Bellman equation.** Under Assumptions 1–2, we define the TD fixed point $\boldsymbol{\theta}^*$ as the unique solution of the projected Bellman equation (Tsitsiklis and Van Roy, 1996), $\boldsymbol{\Phi}\boldsymbol{\theta}^* = \Pi_{\boldsymbol{D}}\, T^\mu\big(\boldsymbol{\Phi}\boldsymbol{\theta}^*\big)$, where $\Pi_{\boldsymbol{D}}$ is the projection operator onto the subspace $\{\boldsymbol{\Phi}\boldsymbol{x} \mid \boldsymbol{x} \in \mathbb{R}^d\}$ with respect to the inner product $\langle \cdot, \cdot \rangle_{\boldsymbol{D}}$.

**Stationary gradient.** Asymptotic behavior of TD(0) is closely tied to the vector field

$$\bar{\boldsymbol{g}}(\boldsymbol{\theta}) := \sum_{s,s'} \pi(s)\, P^\mu(s, s') \left( r(s, s') + \gamma \boldsymbol{\phi}(s')^\top \boldsymbol{\theta} - \boldsymbol{\phi}(s)^\top \boldsymbol{\theta} \right) \boldsymbol{\phi}(s)\,.$$

Moreover, the TD fixed point $\boldsymbol{\theta}^*$ satisfies $\bar{\boldsymbol{g}}(\boldsymbol{\theta}^*) = \boldsymbol{0}$.

---

**Algorithm 1** Projection-free TD(0) with linear function approximation

---
1: **Input:** policy $\mu$, number of iterations $T$, $c \geq 41$
2: $\boldsymbol{\theta}_0 = \mathbf{0}$
3: **for** $t = 0, \ldots, T$ **do**
4:     From $s_t$, sample the action $a_t$ from $\mu$ and move to $s_{t+1}$
5:     $\boldsymbol{g}_t = \left( r(s_t, s_{t+1}) + \gamma \boldsymbol{\phi}(s_{t+1})^T \boldsymbol{\theta}_t - \boldsymbol{\phi}(s_t)^T \boldsymbol{\theta}_t \right) \boldsymbol{\phi}(s_t)$
6:     $\eta_t = \frac{1}{c\,\phi_\infty^2 \log T \log(t+3)\sqrt{t+1}}$
7:     $\boldsymbol{\theta}_{t+1} = \boldsymbol{\theta}_t + \eta_t \boldsymbol{g}_t$
8: **end for**

---

## 4    Projection-free Temporal Difference Learning

In this section, we present our main result: We analyze TD(0) without any projection, as shown in Algorithm 1, and present a robust convergence result for it.

First, we explain what our convergence measure exactly is. In prior work (e.g., Bhandari et al., 2018), the potential function for the convergence analysis was

$$(1 - \gamma) \left\| \boldsymbol{V}_{\boldsymbol{\theta}^*} - \boldsymbol{V}_{\boldsymbol{\theta}} \right\|_{\boldsymbol{D}}^2 \ . \tag{1}$$

Note that when the discount factor $\gamma = 1$, then the potential loses its utility. For this reason, we focus on the better potential function proposed by Liu and Olshevsky (2021):

$$f(\boldsymbol{\theta}) = (1 - \gamma) \left\| \boldsymbol{V}_{\boldsymbol{\theta}} - \boldsymbol{V}_{\boldsymbol{\theta}^*} \right\|_{\boldsymbol{D}}^2 + \gamma \left\| \boldsymbol{V}_{\boldsymbol{\theta}} - \boldsymbol{V}_{\boldsymbol{\theta}^*} \right\|_{\text{Dir}}^2 \ .$$

Clearly, the TD fixed point $\boldsymbol{\theta}^*$ minimizes $f(\boldsymbol{\theta})$. Moreover, any result obtained using this potential implies those obtained using (1). Finally, this potential provides convergence results even when the discount factor $\gamma$ equals 1; see the discussion in Liu and Olshevsky (2021).

From a technical point of view, the advantage of this potential is that the mean-path TD update $\bar{\boldsymbol{g}}(\boldsymbol{\theta})$ satisfies the equality in the following theorem. Instead, using the potential in equation (1), one would obtain an inequality. In a sense, the results in Liu and Olshevsky (2021) indicate that this is the "correct" potential function for TD(0).

**Lemma 4.1.** *(Liu and Olshevsky, 2021, Theorem 1) For any $\boldsymbol{\theta} \in \mathbb{R}^d$, we have*

$$\langle -\bar{\boldsymbol{g}}(\boldsymbol{\theta}), \boldsymbol{\theta} - \boldsymbol{\theta}^* \rangle = f(\boldsymbol{\theta}) - f(\boldsymbol{\theta}^*) \ .$$

Using the above potential function, the following theorem shows that the TD algorithm with a carefully chosen stepsize converges with a rate of $\widetilde{O}(\|\boldsymbol{\theta}^*\|_2^2 / \sqrt{T})$, even in the presence of Markovian noise and without projections.

**Theorem 4.2.** *Suppose we are running Algorithm 1. Let $\bar{\boldsymbol{\theta}}_T \coloneqq \frac{1}{\sum_{i=0}^{T-1} \eta_i} \sum_{k=0}^{T-1} \eta_k \boldsymbol{\theta}_k$. For $T$ sufficiently large depending on $\alpha$ in Theorem 3.1, and $c \geq 41$, we have:*

(a) *For any $t \leq T$, $\mathbb{E}\left[ \|\boldsymbol{\theta}_t\|_2^2 \right] \leq \rho_c^2 \max\left\{ \frac{r_\infty^2}{\phi_\infty^2}, \|\boldsymbol{\theta}^*\|_2^2 \right\}$, where $\rho_c = \mathcal{O}(\frac{1}{(c-41)^2} + 1)$ for both $c \to 41$ and $c \to \infty$ .*

(b) $\mathbb{E}\left[ (1 - \gamma) \left\| \boldsymbol{V}_{\bar{\boldsymbol{\theta}}_T} - \boldsymbol{V}_{\boldsymbol{\theta}^*} \right\|_{\boldsymbol{D}}^2 + \gamma \left\| \boldsymbol{V}_{\bar{\boldsymbol{\theta}}_T} - \boldsymbol{V}_{\boldsymbol{\theta}^*} \right\|_{\text{Dir}}^2 \right] = \widetilde{\mathcal{O}}\left( \frac{c\rho_c^2 \max\{r_\infty^2, \phi_\infty^2 \|\boldsymbol{\theta}^*\|_2^2\}}{\sqrt{T}} \right) .$

We give the proof of part (a) in Theorem F.1; part (b) follows as a corollary at the end of Section 4. Theorem F.1 details how the stepsize scaling parameter $c$ determines the radius multiplier $\rho_c$. In short, $c$ must exceed a threshold to ensure bounded iterates, and increasing $c$ decreases $\rho_c$.

The fact that the iterates are bounded is both a novel contribution and a crucial ingredient in our analysis. We now explain how this is obtained, contrasting it with previous approaches.

**Prior work: Controlling the iterates with strong convexity.** The standard approach to analyze TD(0) starts from the following recursion (Bhandari et al., 2018; Liu and Olshevsky,

2021; Mitra, 2024):

$$\begin{aligned}
\|\boldsymbol{\theta}_t - \boldsymbol{\theta}^*\|_2^2 &= \|\boldsymbol{\theta}_{t-1} + \eta_{t-1}\boldsymbol{g}_{t-1} - \boldsymbol{\theta}^*\|_2^2 \\
&= \|\boldsymbol{\theta}_{t-1} - \boldsymbol{\theta}^*\|_2^2 + 2\eta_{t-1}\langle \boldsymbol{g}_{t-1}, \boldsymbol{\theta}_{t-1} - \boldsymbol{\theta}^* \rangle + \eta_{t-1}^2 \|\boldsymbol{g}_{t-1}\|_2^2 \\
&= \|\boldsymbol{\theta}_{t-1} - \boldsymbol{\theta}^*\|_2^2 + 2\eta_{t-1}\langle \bar{\boldsymbol{g}}(\boldsymbol{\theta}_{t-1}), \boldsymbol{\theta}_{t-1} - \boldsymbol{\theta}^* \rangle + \eta_{t-1}^2 \|\boldsymbol{g}_{t-1}\|_2^2 \\
&\quad + 2\eta_{t-1}\langle \boldsymbol{g}_{t-1} - \bar{\boldsymbol{g}}(\boldsymbol{\theta}_{t-1}), \boldsymbol{\theta}_{t-1} - \boldsymbol{\theta}^* \rangle .
\end{aligned}$$

Then, in the strong convexity setting, one uses the following lemma:

**Lemma 4.3.** *(Mitra, 2024, Lemma 1) Let $\omega > 0$ be the minimal eigenvalue of $\boldsymbol{\Phi}^\top \boldsymbol{D}\boldsymbol{\Phi}$. Then,*

$$\langle \bar{\boldsymbol{g}}(\boldsymbol{\theta}), \boldsymbol{\theta} - \boldsymbol{\theta}^* \rangle \leq -\omega(1-\gamma)\|\boldsymbol{\theta} - \boldsymbol{\theta}^*\|_2^2, \quad \forall \boldsymbol{\theta} \in \mathbb{R}^d .$$

Thus, we have

$$\|\boldsymbol{\theta}_t - \boldsymbol{\theta}^*\|_2^2 \leq (1 - 2\eta_{t-1}\omega(1-\gamma))\|\boldsymbol{\theta}_{t-1} - \boldsymbol{\theta}^*\|_2^2 + \eta_{t-1}^2\|\boldsymbol{g}_{t-1}\|_2^2 + 2\eta_{t-1}\langle \boldsymbol{g}_{t-1} - \bar{\boldsymbol{g}}(\boldsymbol{\theta}_{t-1}), \boldsymbol{\theta}_{t-1} - \boldsymbol{\theta}^* \rangle .$$

Mitra (2024) then proceeds to control the gradient term $\|\boldsymbol{g}_{t-1}\|_2^2$ and bias term $\langle \boldsymbol{g}_{t-1} - \bar{\boldsymbol{g}}(\boldsymbol{\theta}_{t-1}), \boldsymbol{\theta}_{t-1} - \boldsymbol{\theta}^* \rangle$ to form the following standard pseudo-contraction:

$$\mathbb{E}\left[\|\boldsymbol{\theta}_t - \boldsymbol{\theta}^*\|_2^2\right] = (1 - 2\eta\omega(1-\gamma))\mathbb{E}\left[\|\boldsymbol{\theta}_{t-1} - \boldsymbol{\theta}^*\|_2^2\right] + \mathcal{O}(\eta^2\tau\|\boldsymbol{\theta}^*\|_2^2),$$

Mitra (2024) then concludes that $(1-\gamma)\mathbb{E}\left[\|\boldsymbol{V}_{\boldsymbol{\theta}^*} - \boldsymbol{V}_{\bar{\boldsymbol{\theta}}_T}\|_{\boldsymbol{D}}^2\right] = \mathcal{O}(\frac{\tau\|\boldsymbol{\theta}^*\|_2^2}{\omega^2(1-\gamma)T})$ by unrolling the recursion. Other proofs based on strong convexity follow a similar strategy. Notice that there is no need for projection steps in their analysis, thanks to the contraction property, which keeps the iterates bounded. However, in this approach the rate depends on the minimal eigenvalue $\omega$ of $\boldsymbol{\Phi}^\top \boldsymbol{D}\boldsymbol{\Phi}$, which could be arbitrarily small, as we show in Lemma 5.1.

**Prior Work: Controlling the iterates with projections.** To avoid the dependency on $\omega$, we can analyze TD(0) using Lemma 4.1 instead of Lemma 4.3. The analysis starts from controlling the magnitude of the gradient term $\|\boldsymbol{g}_{t-1}\|_2$ using the following lemma:

**Lemma 4.4.** *(Bhandari et al., 2018, Lemma 6) Define $\boldsymbol{g}_t(\boldsymbol{\theta}) := \left(r(s_t, s_{t+1}) + \gamma\phi(s_{t+1})^T\boldsymbol{\theta} - \phi(s_t)^T\boldsymbol{\theta}\right)\phi(s_t)$. Then, for all $\boldsymbol{\theta} \in \mathbb{R}^d$, $\|\boldsymbol{g}_t(\boldsymbol{\theta})\|_2 \leq r_\infty\phi_\infty + 2\phi_\infty^2\|\boldsymbol{\theta}\|_2$.*

Hence, the upper bound on the magnitude of the gradient term $\|\boldsymbol{g}_{t-1}\|_2$ depends on $\|\boldsymbol{\theta}_{t-1}\|_2$. Since $\boldsymbol{\theta}_{t-1} = \boldsymbol{\theta}_{t-2} + \eta_{t-2}\boldsymbol{g}_{t-2}$, the norm $\|\boldsymbol{\theta}_{t-1}\|_2$ in turn depends on both $\|\boldsymbol{\theta}_{t-2}\|_2$ and $\|\boldsymbol{g}_{t-2}\|_2$. This recursive dependence can create a vicious cycle leading to an explosion of $\|\boldsymbol{\theta}_t\|_2$ from the stochasticity of $\boldsymbol{g}_{t-1}$. Previous analyses (Bhandari et al., 2018; Liu and Olshevsky, 2021) avoid this problem by imposing a projection step, which guarantees $\|\boldsymbol{\theta}_t\|_2 \leq R$ for all $t$, where $R$ is chosen agnostically to be larger than $\|\boldsymbol{\theta}^*\|_2$. Under this constraint, we have a uniform control over the magnitude of $\|\boldsymbol{g}_t\|_2$ by $r_\infty\phi_\infty + 2\phi_\infty^2 R$ for all $t$ which can be seen as a bounded gradients condition.

**Our Approach: Controlling the iterates *without* projections.** Now, we explain our proof method that removes the need for a projection. We decompose the updates as follows:

$$\begin{aligned}
\boldsymbol{\theta}_t &= \boldsymbol{\theta}_{t-1} + \eta_t\boldsymbol{g}_{t-1} \\
&= \boldsymbol{\theta}_{t-1} + \eta_t\Big(\underbrace{\boldsymbol{g}_{t-1} - \mathbb{E}[\boldsymbol{g}_{t-1} \mid \mathcal{F}_{t-2}]}_{\boldsymbol{\xi}_{t-1}} + \underbrace{\mathbb{E}[\boldsymbol{g}_{t-1} \mid \mathcal{F}_{t-2}] - \bar{\boldsymbol{g}}(\boldsymbol{\theta}_{t-1})}_{\boldsymbol{b}_{t-1}} + \bar{\boldsymbol{g}}(\boldsymbol{\theta}_{t-1})\Big) .
\end{aligned}$$

Notice that $\boldsymbol{\xi}_{t-1}$ is a martingale difference with respect to $\mathcal{F}_{t-2}$, and $\boldsymbol{b}_{t-1}$ is the gradient bias term. Then, we have

$$\begin{aligned}
\|\boldsymbol{\theta}_t - \boldsymbol{\theta}^*\|_2^2 &= \|\boldsymbol{\theta}_{t-1} + \eta_{t-1}(\boldsymbol{\xi}_{t-1} + \boldsymbol{b}_{t-1} + \bar{\boldsymbol{g}}(\boldsymbol{\theta}_{t-1})) - \boldsymbol{\theta}^*\|_2^2 \\
&= \|\boldsymbol{\theta}_{t-1} - \boldsymbol{\theta}^*\|_2^2 + 2\eta_{t-1}\langle \boldsymbol{\xi}_{t-1} + \boldsymbol{b}_{t-1} + \bar{\boldsymbol{g}}(\boldsymbol{\theta}_{t-1}), \boldsymbol{\theta}_{t-1} - \boldsymbol{\theta}^* \rangle + \eta_{t-1}^2\|\boldsymbol{\xi}_{t-1} + \boldsymbol{b}_{t-1} + \bar{\boldsymbol{g}}(\boldsymbol{\theta}_{t-1})\|_2^2 \\
&\leq \|\boldsymbol{\theta}_{t-1} - \boldsymbol{\theta}^*\|_2^2 + 2\eta_{t-1}\langle \boldsymbol{\xi}_{t-1} + \boldsymbol{b}_{t-1}, \boldsymbol{\theta}_{t-1} - \boldsymbol{\theta}^* \rangle \\
&\quad + 3\eta_{t-1}^2\|\boldsymbol{\xi}_{t-1}\|_2^2 + 3\eta_{t-1}^2\|\boldsymbol{b}_{t-1}\|_2^2 + 3\eta_{t-1}^2\|\bar{\boldsymbol{g}}(\boldsymbol{\theta}_{t-1})\|_2^2 ,
\end{aligned}$$

where we use Lemma 4.1 in the last inequality. Taking expectation and telescoping gives

$$\mathbb{E}\left[\|\boldsymbol{\theta}_t - \boldsymbol{\theta}^*\|_2^2\right] \leq 2\mathbb{E}\left[\sum_{k=0}^{t-1} \eta_k \langle \boldsymbol{b}_k, \boldsymbol{\theta}_k - \boldsymbol{\theta}^* \rangle\right] + 3\mathbb{E}\left[\sum_{k=0}^{t-1} \eta_k^2 (\|\boldsymbol{\xi}_k\|_2^2 + \|\boldsymbol{b}_k\|_2^2 + \|\bar{\boldsymbol{g}}(\boldsymbol{\theta}_k)\|_2^2)\right] + \|\boldsymbol{\theta}^*\|_2^2.$$

Our analysis follows from controlling the bias term $\langle \boldsymbol{b}_k, \boldsymbol{\theta}_k - \boldsymbol{\theta}^* \rangle$ in the update. The hardness of analyzing this term is that the stochastic gradient $\boldsymbol{g}_k$ depends on the past trajectories of the Markov chain, and is not an unbiased estimate of $\bar{\boldsymbol{g}}(\boldsymbol{\theta}_k)$. To be more precise, by defining $Z_k := (s_k, s_{k+1})$ and overloading the notation $\boldsymbol{g}(\boldsymbol{\theta}_k, Z_k) := \left(r(s_k, s_{k+1}) + \gamma \boldsymbol{\phi}(s_{k+1})^T \boldsymbol{\theta}_k - \boldsymbol{\phi}(s_k)^T \boldsymbol{\theta}_k\right) \boldsymbol{\phi}(s_k)$, we have

$$\mathbb{E}_{Z_k \sim (\pi, \pi P^\mu)}[\boldsymbol{g}(\boldsymbol{\theta}_k, Z_k)] \neq \bar{\boldsymbol{g}}(\boldsymbol{\theta}_k),$$

which comes from the fact that $\boldsymbol{\theta}_k$ is also a random variable depending on $\mathcal{F}_{k-1} = \sigma(Z_0, \ldots, Z_{k-1})$, and $Z_k$ depends on $Z_{k-1}$.

The key idea to decouple this dependency comes from the fact that the Markov chain is ergodic: it converges to the stationary distribution geometrically fast, i.e., the chain loses memory of the states earlier than $\tau$ steps back. So, $\boldsymbol{\theta}_{k-\tau}$ can be seen as almost independent of $Z_k$, thanks to the following lemma.

**Lemma 4.5.** *(Bhandari et al., 2018, Lemma 9) Let $\ell_{k-\tau} := r_\infty \phi_\infty + 2\phi_\infty^2 \|\boldsymbol{\theta}_{k-\tau}\|_2$, then we have*

$$\|\mathbb{E}[\boldsymbol{g}(\boldsymbol{\theta}_{k-\tau}, Z_k) - \bar{\boldsymbol{g}}(\boldsymbol{\theta}_{k-\tau}) \mid \mathcal{F}_{k-\tau-1}]\|_2 \leq 2\ell_{k-\tau} C \alpha^\tau .$$

With this observation, we can decompose the bias term as

$$\mathbb{E}[\langle \boldsymbol{g}_k - \bar{\boldsymbol{g}}(\boldsymbol{\theta}_k), \boldsymbol{\theta}_k - \boldsymbol{\theta}^* \rangle \mid \mathcal{F}_{k-\tau-1}] = \mathbb{E}[\langle \boldsymbol{g}(\boldsymbol{\theta}_{k-\tau}, Z_k) - \bar{\boldsymbol{g}}(\boldsymbol{\theta}_{k-\tau}), \boldsymbol{\theta}_k - \boldsymbol{\theta}^* \rangle \mid \mathcal{F}_{k-\tau-1}]$$
$$+ \mathbb{E}[\langle \boldsymbol{g}(\boldsymbol{\theta}_k, Z_k) - \boldsymbol{g}(\boldsymbol{\theta}_{k-\tau}, Z_k), \boldsymbol{\theta}_k - \boldsymbol{\theta}^* \rangle \mid \mathcal{F}_{k-\tau-1}] + \mathbb{E}[\langle \bar{\boldsymbol{g}}(\boldsymbol{\theta}_{k-\tau}) - \bar{\boldsymbol{g}}(\boldsymbol{\theta}_k), \boldsymbol{\theta}_k - \boldsymbol{\theta}^* \rangle \mid \mathcal{F}_{k-\tau-1}]$$

Hence, we can control the bias term by combining Lemma 4.5 and Cauchy inequality for the first line of the right hand side, and for the second line with the following lemma that shows that $\boldsymbol{g}(\boldsymbol{\theta}, Z_k)$ and $\bar{\boldsymbol{g}}(\boldsymbol{\theta})$ are Lipschitz in $\boldsymbol{\theta}$.

**Lemma 4.6.** *(Bhandari et al., 2018, Lemma 10) Fix any $k \leq T$, for any $Z_k$, $\boldsymbol{\theta}$ and $\boldsymbol{\theta}'$, we have*

$$\|\boldsymbol{g}(\boldsymbol{\theta}, Z_k) - \boldsymbol{g}(\boldsymbol{\theta}', Z_k)\|_2 \leq 2\phi_\infty^2 \|\boldsymbol{\theta} - \boldsymbol{\theta}'\|_2 \quad \text{and} \quad \|\bar{\boldsymbol{g}}(\boldsymbol{\theta}) - \bar{\boldsymbol{g}}(\boldsymbol{\theta}')\|_2 \leq 2\phi_\infty^2 \|\boldsymbol{\theta} - \boldsymbol{\theta}'\|_2 .$$

Notice that $\mathbb{E}\left[\sum_{k=0}^{t-1} \eta_k \langle \boldsymbol{b}_k, \boldsymbol{\theta}_k - \boldsymbol{\theta}^* \rangle\right] = \mathcal{O}(\max_{i \leq t-1} \mathbb{E}\left[\|\boldsymbol{\theta}_i\|_2^2\right] + \|\theta^*\|_2^2)$ from previous lemmas and $\|\boldsymbol{\theta}_k - \boldsymbol{\theta}^*\|_2 \leq \|\boldsymbol{\theta}_k\|_2 + \|\boldsymbol{\theta}^*\|_2$.

It follows that $\mathbb{E}\left[\|\boldsymbol{\theta}_t - \boldsymbol{\theta}^*\|_2^2\right] = \mathcal{O}(\max_{i \leq t-1} \mathbb{E}\left[\|\boldsymbol{\theta}_i\|_2^2\right] + \|\boldsymbol{\theta}^*\|_2^2)$. Notice that $\mathbb{E}[\|\boldsymbol{\theta}_t\|_2^2] \leq \mathbb{E}[(\|\boldsymbol{\theta}_t - \boldsymbol{\theta}^*\|_2 + \|\boldsymbol{\theta}^*\|_2)^2]$ from the triangular inequality, so $\mathbb{E}\left[\|\boldsymbol{\theta}_t - \boldsymbol{\theta}^*\|_2^2\right]$ and $\mathbb{E}\left[\|\boldsymbol{\theta}_t\|_2^2\right]$ have a tangled recursion structure. Reasoning by induction, we assume

$$\max_{i \leq t-1} \mathbb{E}\left[\|\boldsymbol{\theta}_i\|_2^2\right] \leq \rho_c^2 \max\left\{\frac{r_\infty^2}{\phi_\infty^2}, \|\boldsymbol{\theta}^*\|_2^2\right\},$$

and aim for showing $\mathbb{E}\left[\|\boldsymbol{\theta}_t\|_2^2\right] \leq \rho_c^2 \max\left\{\frac{r_\infty^2}{\phi_\infty^2}, \|\boldsymbol{\theta}^*\|_2^2\right\}$ with the help of the gateway $\mathbb{E}\left[\|\boldsymbol{\theta}_t - \boldsymbol{\theta}^*\|_2^2\right]$. In words, if the stepsize is small enough, if a constant bounds the previous iterates, the next one will be bounded by the same constant. Indeed, for some $\beta > 0$, we have

$$\mathbb{E}\left[\|\boldsymbol{\theta}_t\|_2^2\right] \leq \mathbb{E}\left[(\|\boldsymbol{\theta}_t - \boldsymbol{\theta}^*\|_2 + \|\boldsymbol{\theta}^*\|_2)^2\right] \leq \left(\|\boldsymbol{\theta}^*\|_2 + \sqrt{\mathbb{E}\left[\|\boldsymbol{\theta}_t - \boldsymbol{\theta}^*\|_2^2\right]}\right)^2$$

$$\leq \left(\|\boldsymbol{\theta}^*\|_2 + \beta \sqrt{\max_{i \leq t-1} \mathbb{E}\left[\|\boldsymbol{\theta}_i\|_2^2\right]}\right)^2 \leq \rho_c^2 \max\left\{\frac{r_\infty^2}{\phi_\infty^2}, \|\boldsymbol{\theta}^*\|_2^2\right\},$$

where we use the induction hypothesis in the last equality and we remark again that the stepsize $c$ and $\rho_c$ are chosen carefully to ensure the correctness of the induction proof. The precise statement and proof can be found in Theorem F.1 in the Appendix.

**Convergence result.** We now give the proof of Theorem 4.2 (b).

*Proof.* For any $0 < t \leq T$, let $d_t = \|\boldsymbol{\theta}^* - \boldsymbol{\theta}_t\|_2$. We have

$$
\begin{aligned}
d_t^2 &= \|\boldsymbol{\theta}^* - \boldsymbol{\theta}_{t-1} - \eta_{t-1}\boldsymbol{g}_{t-1}\|_2^2 \\
&= d_{t-1}^2 - 2\eta_{t-1}\langle\boldsymbol{g}_{t-1}, \boldsymbol{\theta}^* - \boldsymbol{\theta}_{t-1}\rangle + \eta_{t-1}^2\|\boldsymbol{g}_{t-1}\|_2^2 \\
&= d_{t-1}^2 - 2\eta_{t-1}\langle\bar{\boldsymbol{g}}(\boldsymbol{\theta}_{t-1}), \boldsymbol{\theta}^* - \boldsymbol{\theta}_{t-1}\rangle + 2\eta_{t-1}\langle\bar{\boldsymbol{g}}(\boldsymbol{\theta}_{t-1}) - \boldsymbol{g}_{t-1}, \boldsymbol{\theta}^* - \boldsymbol{\theta}_{t-1}\rangle + \eta_{t-1}^2\|\boldsymbol{g}_{t-1}\|_2^2 \ .
\end{aligned}
$$

Summing from $t = 0$ to $t = T - 1$, taking the expectation, and using Lemma 4.1, we have

$$
\begin{aligned}
\sum_{t=0}^{T-1} 2\eta_t &\mathbb{E}\Big[(1 - \gamma)\|\boldsymbol{V}_{\boldsymbol{\theta}_t} - \boldsymbol{V}_{\boldsymbol{\theta}^*}\|_{\boldsymbol{D}}^2 + \gamma\|\boldsymbol{V}_{\boldsymbol{\theta}_t} - \boldsymbol{V}_{\boldsymbol{\theta}^*}\|_{\mathrm{Dir}}^2\Big] \\
&\leq \sum_{t=0}^{T-1}\left(\mathbb{E}\big[d_t^2\big] - \mathbb{E}\big[d_{t+1}^2\big]\right) + \mathbb{E}\left[\sum_{t=0}^{T-1} 2\eta_t\langle\bar{\boldsymbol{g}}(\boldsymbol{\theta}_t) - \boldsymbol{g}_t, \boldsymbol{\theta}^* - \boldsymbol{\theta}_t\rangle\right] + \mathbb{E}\left[\sum_{t=0}^{T-1}\eta_t^2\|\boldsymbol{g}_t\|_2^2\right] \\
&= \|\boldsymbol{\theta}^*\|_2^2 + \mathcal{O}\left(\rho_c^2\max\left\{\frac{r_\infty^2}{\phi_\infty^2}, \|\boldsymbol{\theta}^*\|_2^2\right\}\right),
\end{aligned}
$$

where we used Theorem F.1 in the last inequality. Using the convexity of $f$, we have

$$
\begin{aligned}
\mathbb{E}\big[f(\bar{\boldsymbol{\theta}}_T) - f(\boldsymbol{\theta}^*)\big] &\leq \frac{1}{\sum_{i=0}^{T-1}\eta_i}\sum_{t=0}^{T-1}\eta_t\mathbb{E}\Big[(1 - \gamma)\|\boldsymbol{V}_{\boldsymbol{\theta}_t} - \boldsymbol{V}_{\boldsymbol{\theta}^*}\|_{\boldsymbol{D}}^2 + \gamma\|\boldsymbol{V}_{\boldsymbol{\theta}_t} - \boldsymbol{V}_{\boldsymbol{\theta}^*}\|_{\mathrm{Dir}}^2\Big] \\
&= \widetilde{\mathcal{O}}\left(\frac{c\rho_c^2\max\{r_\infty^2, \phi_\infty^2\|\boldsymbol{\theta}^*\|_2^2\}}{\sqrt{T}}\right),
\end{aligned}
$$

where the last equality follows from $\sum_{i=0}^{T-1}\eta_i \geq \sum_{i=0}^{T-1}\frac{1}{c\phi_\infty^2\log^2(T+3)\sqrt{t+1}} \geq \frac{2\sqrt{T}-2}{c\phi_\infty^2\log^2(T+3)}$. $\qquad\square$

**Is the threshold on the stepsize real?** Theorem 4.2 gives a sufficient condition on $c$ to have bounded iterates and convergence. However, one might wonder if this condition is necessary too. That is, does TD(0) without projection have bounded iterates with arbitrary stepsizes? To test this effect, we conducted an experiment in which we ran TD(0) on a synthetic problem (details in Appendix H). In Figure 1, first column, we show the expected boundedness ratio, defined as $\frac{\max_{i \leq T-1}\mathbb{E}\big[\|\boldsymbol{\theta}_i\|_2^2\big]}{\|\boldsymbol{\theta}^*\|_2^2}$, which is large if the iterates blow up. The second column shows the divergence rate, that is, the fraction of runs with $\|\boldsymbol{\theta}_t\|_2^2 > 10^{12}$, while the third column shows the suboptimality gap. Overall, we have that the threshold on the stepsize is indeed real. In fact, from both the expected boundedness ratio and the divergence rate, it is clear that if $c$ is too small, the iterates of the algorithm are not controlled. Moreover, the explosion in both of these measures nicely mirrors the theoretical behavior in our function $\rho_c$ in Theorem 4.2.

The different rows represent different spectral characteristics of $\boldsymbol{\Phi}^\top\boldsymbol{D}\boldsymbol{\Phi}$. Hence, changing the spectral characteristics does not change much the iterates, as our theory suggests. However, it does influence the suboptimality gap, suggesting that a robust rate of $\widetilde{\mathcal{O}}(1/\sqrt{T})$ might be pessimistic when the strong convexity is large.

In Appendix H we also report experiments with a fixed stepsize showing the same behaviors.

## 5 Detailed Comparison with Previous Results and Limitations

In this section, we discuss some technical differences between our results and previous ones. Let's start with Bhandari et al. (2018). Their Theorem 3 proves that projected TD satisfies

$$
(1 - \gamma)\mathbb{E}\Big[\big\|\boldsymbol{V}_{\bar{\boldsymbol{\theta}}_T} - \boldsymbol{V}_{\boldsymbol{\theta}^*}\big\|_{\boldsymbol{D}}^2\Big] = \widetilde{\mathcal{O}}\left(\frac{R^2}{\sqrt{T}}\right),
$$

where $R$ is a parameter of the algorithm and it must be chosen to satisfy $R \geq \|\boldsymbol{\theta}^*\|_2$, where $\|\boldsymbol{\theta}^*\|_2$ is unknown. Our result recovers the above bound since $\|\boldsymbol{V}_{\boldsymbol{\theta}} - \boldsymbol{V}_{\boldsymbol{\theta}^*}\|_{\mathrm{Dir}}^2 \geq 0$, but, given that we do not have projections, we depend directly on $\|\boldsymbol{\theta}^*\|_2$ instead of $R$. While in

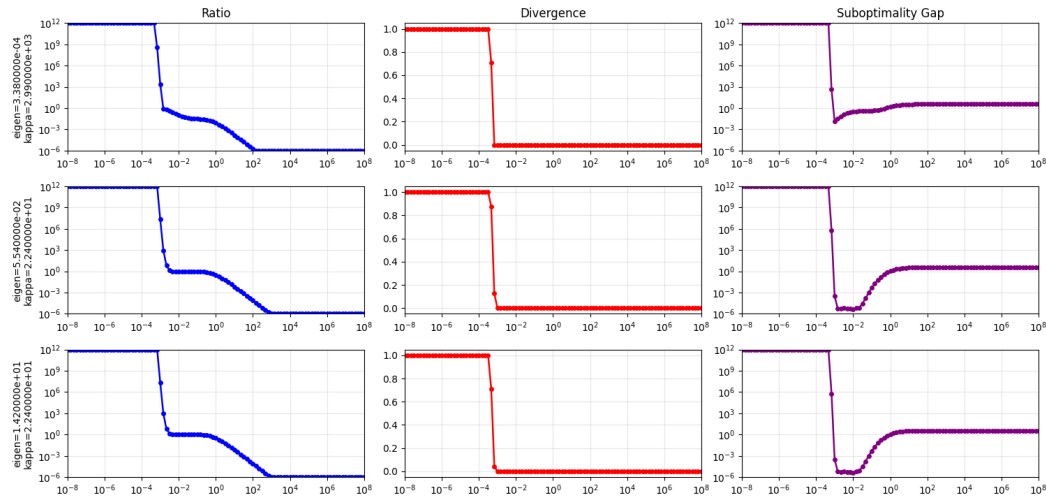

Figure 1: Instability of TD learning. Columns: boundedness ratio, divergence rate, suboptimality gap vs. stepsize scale (c). Rows: different feature scalings (changing the spectrum of $\mathbf{\Phi}^\top \boldsymbol{D}\mathbf{\Phi}$). "Eigen" = minimum eigenvalue, "kappa" = condition number.

principle $\|\boldsymbol{\theta}^*\|_2$ could be estimated using $\omega$ (Bhandari et al., 2018, Lemma 1), this approach is impractical from an algorithm design perspective. In fact, the goal of TD learning is precisely to avoid the computational complexity depending on the number of states $n$, which is highly non-trivial when estimating $\omega$ involves estimating the stationary matrix $D$.

For additional clarity, we also calculate the curvature of our potential function and show that it could be arbitrarily small (proof in Appendix G).

**Lemma 5.1.** *Suppose $\|\boldsymbol{\phi}\|_2 \leq \phi_\infty$ and $span(\{\boldsymbol{\phi}(s_1), \ldots, \boldsymbol{\phi}(s_n)\}) = \mathbb{R}^d$. Then, we have $\nabla^2 f = 2\mathbf{\Phi}^\top[(1-\gamma)\boldsymbol{D} + \gamma\boldsymbol{L}]\mathbf{\Phi}$. Moreover, depending on the features $\mathbf{\Phi}$, the minimum eigenvalue of the Hessian can be arbitrarily small.*

A limitation of our result is that we require $T$ to be large enough to eliminate dependence on the mixing parameter $\alpha$ in Theorem 3.1. However, this is a common limitation in the literature, sometimes hidden. In fact, either the known bounds are truly of order $\widetilde{\mathcal{O}}(1/\sqrt{T})$ only when $\log T \geq \tau(1/\sqrt{T})$ (see Appendix I for a proof), or for a sufficiently small stepsize that depends on $\omega$. There are $\omega$-agnostic algorithms in the fast regime (e.g. Patil et al., 2023; Samsonov et al., 2024), but they rely on data dropping that still requires knowledge of the mixing time $\tau$ and $T$, see Table 1. This limitation might be unavoidable as different policies induce different transition kernels and so different mixing behaviors (Nagaraj et al., 2020). Moreover, knowing $T$ in advance per se is not a limitation because one can use a doubling trick, see Appendix J.

## 6 CONCLUSION

In this paper, we present a robust finite-time analysis of TD(0) without requiring additional projection steps. To the best of our knowledge, this is the first finite-time guarantee in this setting. In particular, we do not employ the contraction-based proof technique used in previous work, but instead we directly prove that the iterates of TD(0) are bounded. We believe our proof is general and, for example, it can be easily extended to the TD($\lambda$) and Q-learning setting (Bhandari et al., 2018; Chen et al., 2022).

In future work, we plan to investigate the possibility of obtaining rates that interpolate between $\widetilde{\mathcal{O}}(1/\sqrt{T})$ and $\widetilde{\mathcal{O}}(1/T)$, depending on the curvature of the potential function. Ideally, one would like to show that TD(0) adapts to the curvature of the function with a specific stepsize, as is possible with recent parameter-free schemes (Cutkosky and Orabona, 2018).

## 7 Ethics Statement

We adhere to the ICLR code of Ethics, and this work is theoretical in nature and does not involve direct societal applications. However, foundational advances in optimization theory can indirectly impact a wide range of fields, including machine learning, data science, and operations research.

## 8 Reproducibility Statement

We provide a detailed proof of our main theorem in the appendix. All the lemmas and technical results used in the proof are also provided in the appendix. We also discuss the limitations at the end of Section 5. The code for the experiments is available at https://anonymous.4open.science/r/ICLR2026TD-D12465.

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
