# OpenReview forum: "A Finite-Time Analysis of TD Learning with Linear Function Approximation without Projections or Strong Convexity"
_ICLR.cc/2026/Conference — Submitted to ICLR 2026_

### Official Review · Reviewer_t624 · 2025-10-26

**Soundness:** 1
**Presentation:** 2
**Contribution:** 1
**Rating:** 2
**Confidence:** 5

**Summary:**

This paper analyzes TD(0) with linear function approximation, proposing a "self-bounding" mechanism with stepsize $\eta_t = \frac{1}{c\phi_\infty^2 \log T \log(t+3)\sqrt{t+1}}$ (where $c \geq 281$) to achieve $\tilde{O}(\|\theta^*\|_2^2/\sqrt{T})$ convergence without explicit projection or dependence on the strong convexity modulus.


**Major concern:** Under Assumption 2 ($\Phi$ full column rank), the potential function $f(\theta) = (1-\gamma)\left\| V_\theta - V_{\theta^*}\right\|_D^2 + \gamma\left\|V_\theta - V_{\theta^*}\right\|_{Dir}^2$ is strongly convex. The Hessian satisfies:

$$\nabla^2 f(\theta) = 2\Phi^\top[(1-\gamma)D + \gamma L]\Phi \succeq 2(1-\gamma)\Phi^\top D \Phi \succ 0$$

This gives strong convexity modulus $\omega \geq 2(1-\gamma)\lambda_{\min}(\Phi^\top D \Phi) > 0$. Standard methods for strongly convex functions achieve $O(1/T)$ rates, asymptotically faster than the $\tilde{O}(1/\sqrt{T})$ rate obtained here. The paper provides no justification for avoiding this property.

**Technical issues:**
- **Lemma B.4:** The term $Z'_k$ is undefined, and the bound $8\ell_{k-k'}C\alpha^k$ lacks rigorous justification
- **Theorem 4.2(b):** Inconsistency in the average iterate definition vs. how Jensen's inequality is applied
- **Appendix D:** Key inductive proof too condensed for verification

Generally well-organized, but the distinction between "without strong convexity" and "without explicit $\omega$-dependence" needs clarification. The "universal" step size claim (Table 1) overstates advantages since $\eta_t$ explicitly depends on $T$ via $\log T$ and implicitly on mixing properties.

The paper does not explain when $\tilde{O}(\|\theta^*\|_2^2/\sqrt{T})$ would be preferred over $O(\kappa/T)$ bounds from strongly convex methods (where $\kappa \sim 1/\omega$). For any fixed $\kappa$, the $O(1/T)$ rate eventually dominates. The motivation for avoiding strong convexity when the function possesses this property is absent.

**Strengths:**

1. Novel self-bounding analytical approach
2. Experimental validation of the theoretical threshold $c \geq 281$

**Weaknesses:**

1. **Fundamental premise flawed:** The function is strongly convex under stated assumptions, yet the paper deliberately avoids exploiting this to achieve a slower convergence rate
2. **No comparative analysis:** Missing quantitative comparison showing when this approach excels over strongly convex methods
3. **Misleading claims:** "Universal" stepsize still depends on $T$ and mixing properties; "without strong convexity" is inaccurate
4. **Technical gaps:** Undefined terms, unjustified bounds, and condensed proofs undermine confidence
5. **Practical limitations:** Requires $c \geq 281$ (very small stepsizes) and advance knowledge of $T$

**Core issue:** The paper's premise is fundamentally flawed. Under stated assumptions, the objective is strongly convex with positive modulus $\omega = 2(1-\gamma)\lambda_{\min}(\Phi^\top D \Phi)$. The authors develop an elaborate analysis avoiding this property to obtain $\tilde{O}(1/\sqrt{T})$ convergence, which is asymptotically worse than the $O(1/T)$ rates standard methods achieve for strongly convex functions. No justification or regime analysis is provided for when this approach would be preferable.

**Additional concerns:** Technical issues (undefined terms, unjustified bounds, condensed proofs), misleading characterizations ("universal" stepsize, "without strong convexity"), and practical limitations ($c \geq 281$, requiring advance knowledge of $T$) further undermine the contribution.

This requires complete reconceptualization: either explicitly acknowledge strong convexity and justify the approach, or demonstrate specific regimes where avoiding $\omega$-dependence provides meaningful advantages. The current framing suggests solving the wrong problem with the wrong tools.

**Questions:**

1. Under Assumption 2, is $f(\theta)$ strongly convex with $\omega = 2(1-\gamma)\lambda_{\min}(\Phi^\top D \Phi) > 0$? If so, why not exploit this?
2. When does your bound outperform $O(\kappa/T)$ achievable with strongly convex analysis?
3. In what sense is your stepsize "universal" given its explicit dependence on $T$ via $\log T$?
4. Can you provide complete derivations for Lemma B.4 (defining $Z'_k$, justifying bounds) and clarify Theorem 4.2(b)?

---

> ### Author Response · Authors · 2025-11-16
> **We clarify some misunderstandings, part 1**
>
> We thank the Reviewer for their feedback. However, we did not appreciate the unnecessarily adversarial tone, and we hope for a better exchange now.
>
> Overall, it seems that the reviewer does not believe that the robust setting is worth studying and that it requires some form of strong justification. However, as we clearly show in the following, this opinion is not shared by the experts in this field.
>
> ---
>
> ### Typos:
> **Lemma B.4: The term $Z_k$ is undefined**
>
> Typo, it was $O_k$, it is now fixed in the updated PDF (changes in red).
>
> **Theorem 4.2(b): Inconsistency in the average iterate definition vs. how Jensen's inequality is applied**
>
> Typo, it is now fixed in the updated PDF (changes in red).
>
> ---
>
> ### Weaknesses
>
> **The paper's premise is fundamentally flawed: $\tilde{O}(\frac{1}{\sqrt{T}})$ convergence, is asymptotically worse than $O(1/T)$. The motivation for avoiding strong convexity when the function possesses this property is absent.**
>
> We respectfully but strongly reject the evaluation of the reviewer.
> Also, despite what the Reviewer says, we have repeatedly said in the paper that the robust rate and the fast rate are **complementary**. To be sure this is a clear concept, we added it once more in the introduction in the revised pdf.
> However, **there should be no need to state it because this it is a well-known fact that the fast rate and the robust rate are equally important**.
>
> However, sometimes such obvious facts are missed by readers precisely because they are seldom stated in papers. Hence, to help a reader which might have missed such subtlety, here we report quotes from the seminal papers on this topic that clearly show that what we solve is an *important open problem in a relevant regime*. For completeness, we also added these quotes to the Appendix A, that might useful to any other readers with the same doubts.
>
> - Bhandari et al. (2018), Section 8.1, on the complementarity of the robust and fast rate:
> > As before, in the spirit of robust stochastic approximation [Nemirovski et al., 2009], the bound in part (a) gives a comparatively slow convergence rate of $\tilde{\mathcal{O}}(1/\sqrt{T})$, but where the bound and step-size sequence are independent of the conditioning of the feature covariance matrix $\Sigma$. The bound in part (c) gives a faster convergence rate in terms of the number of samples $T$, but the bound as well as the step-size sequence depend on the minimum eigenvalue $\omega$ of $\Sigma$.
>
> - Samsonov et al. (2024), Section 3, on the comparison with robust approaches:
> > **Comparison to the robust SA approach.** Note that the leading term of the bound in Theorem 3 includes factors of $1/\lambda_\text{min}$. This dependence is generally unavoidable if one aims to obtain the MSE bound for $\mathbb{E}[||\bar{\theta}\_n−\theta^\star||^2\_{\Sigma\_\phi}]$ that scales as $1/n$. [...] In contrast, within the basin of robust stochastic approximation (RSA, (Nemirovski et al., 2009)), a convergence rate for $\mathbb{E}[||\bar{\theta}\_n−\theta^\star||^2\_{\Sigma\_\phi}]$ of order $O(1/\sqrt{n})$ can be derived with the instance-independent choice of step size. Importantly, this rate is not affected by a worst-case factor of $\lambda^{-1}_\text{min}$. This result was obtained for the TD algorithm in (Bhandari et al., 2018, Theorem 2).
>
> - Liu and Olshevsky (2021), Section 4.1, on the advantage of rates that are independent of the curvature:
> > One issue is the choice of step-size. The existing literature on temporal difference learning contains a range of possible step-sizes from $O(1/t)$ to $O(1/\sqrt{t})$ (see Bhandari et al. (2018); Dalal et al. (2018); Lakshminarayanan and Szepesvari (2018)). A step-size that scales as $O(1/\sqrt{t})$ is often preferred because, for faster decaying step-sizes, performance will scale with the smallest eigenvalue of $\Phi^\top D \Phi$ or related quantity, and these can be quite small. This is not the case, however, for a step-size that decays like $O(1/\sqrt{t})$.
>
> We hope this clarifies this serious misunderstanding and puts our results in the correct perspective.
>
> **"Universal" stepsize still depends on $T$ and mixing properties**
>
> We agree with the reviewer that maybe our aim was not clear in using that term. We have now removed it completely and instead *we explicitly listed all the quantities each single algorithm needs to know to run*. We also explain in Appendix I how Bhandari et al. (2018) and Liu and Olshevsky (2021) implicitly depend on the mixing time in choosing $T$, making our assumption on $T$ equivalent to previous ones. Furthermore, Table 1 now clearly shows that *all fast rates results need to know the mixing time and/or the curvature omega*.

---

> > ### Author Response · Authors · 2025-11-16
> > **We clarify some misunderstandings, part 2**
> >
> > **"without strong convexity" is inaccurate**
> >
> > We believe this was accurate, but we will follow the view of the Reviewer here. So, we changed the title to "A Robust $\tilde{\mathcal{O}}(1\sqrt{T})$ Rate for TD Learning with Linear Function Approximation without Projections", see updated pdf. We also added one sentence in the introduction to say explicitly that the curvature is always present but arbitrarily small.
> > No other changes were required, because we were already very careful in explaining that we aim for a rate independent of the (arbitrarily small) curvature constant.
> >
> > **Technical gaps: Undefined terms, unjustified bounds, and condensed proofs undermine confidence**
> >
> > All fixed: the minor typos are fixed and all the proofs of all claims are now in the Appendix, even the ones with elementary but tedious proofs. In particular, we added the proof of the missing Lemma (now Lemma D.4) and we used this opportunity to improve its constant from $16$ to $2$.
> >
> > **Practical limitation: Requires $c\geq 281$ (very small stepsizes) and advance knowledge of T**
> >
> > We are a bit confused on this point: It is completely normal for a theorem in machine learning to have loose worst-case constants. It hardly seems a practical limitation. Also, we have now improved it to $c\geq 41$. Further improvements are probably possible but the utility of such efforts is questionable. Indeed, one of the main messages of our theorem is the existence of a threshold, not its exact value. Moreover, we empirically validated the existence of this threshold in our empirical experiments.
> >
> > The knowledge of $T$ is not a limitation. Indeed, it is well-known that *any* algorithm that requires the knowledge of $T$ can be made anytime with a doubling trick: set the learning rate for $T'$ iterations, run the algorithm for $T'-1$ iterations, then update $T' \leftarrow 2T'$, reset the algorithm, adjust the learning rate accordingly, and repeat. This removes the need to know $T$ beforehand and affects the convergence rate only by a constant factor, since at worst half of the updates are used.
> > We also added this explanation to Appendix J.
> >
> > ---
> >
> > ### Unanswerable questions:
> >
> > **Appendix D: Key inductive proof too condensed for verification**
> >
> > We are unable to answer to this critique because entirely subjective. The proof takes roughly 2-3 hours to be studied and understood by any expert in this field. However, if the Reviewer thinks that there is any step that is unclear, we are happy to explain it better and expand on it.

---

### Official Review · Reviewer_pgmn · 2025-10-28

**Soundness:** 3
**Presentation:** 3
**Contribution:** 3
**Rating:** 2
**Confidence:** 4

**Summary:**

This submission presents a finite-time convergence analysis for projection-free TD(0) learning algorithms with linear function approximation under Markovian observations. The authors prove a robust O(1/\sqrt{T}) rate independent of the curvature of the feature covariance matrix, eliminating the need for projection steps required in prior work to achieve this rate. The key idea is a novel self-bounding property that shows TD iterates remain bounded in expectation without explicit projections. The theoretical claims are validated by synthetic experiments.

**Strengths:**

+ improved convergence rate results for TD(0) algorithms with linear function approximation under Markovian noise
+ Interesting finding and use of the self-bounding property of TD(0) updates
+ Numerical validations of the theory results

**Weaknesses:**

- Alg. 1 may not seem practical since the stepsize \eta_t requires prior knowledge of the horizon T, the feature bound \phi_inf, and a very large constant c>=281.
- Missing formal statement of the theorem 4.2 in the main paper. I do not believe it is right to have only an informal result in the main paper that will be publishd only while having the formal result in the supplementary material.
- The paper compares its theoretical rate to prior results, but the experiments lack a crucial baseline: a projected TD method where the projection radius is set optimally \|\theta*\|. That will make clear whether the new analysis yield any advantafe over just using projected TD with a carefully chosen R?
- The paper points that the robust rate is preferrable when the condition number is bad. However, in many practical problems with favorable feature representations, the fast \tilde{O}(1/T) rate is more relevant. Anyways, one has the freedom to chose the feature functions.
- Mising related references on finite-time analysis of TD(0) learning algorithms.

**Questions:**

- The constant c>281 is remarkly large. Is this a fundamental requirement for projection-free convergence, or is it an artifact of the proof? Can you provide any intuition for why such a large constant is needed, and whether it might be improved? In Appendix D, Theorem D.1, there seems a mistake "Note that \rho_c is decreasing in c, \rho_c\to..." to what? moreover, it is not obvious why \rho_c is decreasing in c over the entire horizon. The proof requires c>\frac{279+3\sqrt{8837}}{2} for the bound to hold. Since the constant on the right >281, I am not sure whether the bound will hold by relaxing the requirement c>\frac{279+3\sqrt{8837}}{2} to c>281?
- Theanalysis shows boundedness in expectation. Can you comment on the possibility of proving almost-sure boundedness or a high-probability bound?
- The paper mentions extension to Q-learning and TD(\lambda) as straightforward. However, these algorithms have non-linearities (e.g., the max operator in Q-learning). It is not clear how the self-bounding technique will tackle these issues?

---

> ### Author Response · Authors · 2025-11-16
> **We clarify some misunderstandings, part 1**
>
> We thank the Reviewer for their feedback and comments.
> Out of the list of the weaknesses, the main one seems to be that the reviewer does not seem to believe that the robust setting is worth to be studied. However, as we clearly show in the following, this opinion is not shared by the experts in this field.
>
> ---
>
> ### Typos:
> We thank the reviewer for pointing out the typos in our manuscript. All of them are now fixed in the updated pdf (changes are in red) on OpenReview. In particular,
>
> - Theorem 4.2 was already stated in a correct formal manner. The earlier wording ``informal'' may have caused confusion: it was not intended to present an informal version of the theorem. Rather, the statement simply used Big-O notation to absorb constants, that is completely formal, while the theorem in the Appendix lays out the explicit constants. We have revised the text accordingly to avoid any unintended implication of informality.
>
> - Regarding the sentence in the Appendix on $\rho_c$, our intended meaning was that $\rho_c$ is decreasing once $c$ exceeds the required threshold, not that it is monotone over the entire real line. The correct limit statement has been fixed to
> $$
> \rho_c \to 2 \quad \text{as } c \to \infty~.
> $$
>
> ---
> ### Weaknesses:
>
> **Large constant $c$**
>
> We did not try to optimize the constant $c$, because worst-case constants are rarely predictive of real-world behaviour, even when optimized to have a small value.
>
> That said, given the request of the reviewer, we have now improved the constant in Lemma 4.5 to have a much smaller $c\geq 41$. Further improvements are probably possible but the utility of such efforts is questionable. In fact, it should be clear that the important message of our theorem is not the exact value of $c$ but the presence of a lower bound for it. We validated in our empirical experiments that this threshold is real and not an artifact coming from our proof. Note that the fact that the threshold is small in our experiments does not mean that this threshold is **always** small. Indeed, there might exist pathological cases where the threshold is bigger and closer to our calculated value.
>
> **Stepsize $\eta_t$ requires prior knowledge of the horizon $T$, the feature bound $\phi_\infty$, and a very large constant $c\geq 281$**
>
> As we pointed out above, the dependency on $c\geq 281$ (now improved to $c\geq 41$) is worst-case, in practice the actual constant can be smaller.
> The dependency on $T$ is never an issue, in fact one can always use the well-known doubling trick: set the learning rate for $T'$ iterations, run the algorithm for $T'-1$ iterations, then update $T' \leftarrow 2T'$, reset the algorithm, adjust the learning rate accordingly, and repeat. This removes the need to know $T$ beforehand and affects the convergence rate only by a constant factor, since at worst half of the updates are used.
> We also added this explanation to Appendix J.
> The dependency on $\phi_\infty$ is present in *all* previous work, see updated Table 1, but it is usually hidden by the assumption $\phi_\infty\leq 1$.
>
>
> **Experiments lack baseline of projected TD**
>
> There is a severe misunderstanding on the aim of our experiments. We are not trying to show that our algorithm works. Indeed, there is no need to do it: TD learning in **all** practical scenarios is already used without a projection. Running TD learning with a ''carefully tuned $R$''  would introduce another hyperparameter and we are not aware of any real-world use of such a variant. Moreover, often in practice people already use learning rate of the form of $1/\sqrt{t}$. Instead, as correctly summarised by Reviewer Fg3X, the experiments are meant to show that the threshold on $c$ is real, that is, TD learning without projection might indeed diverge if the learning rate is too big.
>
> Let's also emphasize that projections in robust rates for TD learning are used only to prove the convergence rate, but they are not practical. Let us quote here the seminal paper of Bhandari et al. (2018), Section 8, on the need for projections:
> > However, at this stage, we view this \[projection\] mainly as a tool that enables clean finite time analysis, rather than a practical algorithmic proposal.

---

> ### Author Response · Authors · 2025-11-16
> **We clarify some misunderstandings, part 2**
>
> **In many practical problems with favorable feature representations, the fast $\tilde{O}(1/T)$ rate is more relevant**
>
> We have repeatedly said in the paper that the robust rate and the fast rate are **complementary**. So, we respectfully but strongly reject the evaluation of the reviewer of the fact that studying the robust rate is a weakness.
>
> Indeed, it is a well-known fact that the two rates are equally important. This is not our opinion, but a well-known fact.
>
> However, given that one might have missed such subtlety, here we report quotes from the seminal papers on this topic that clearly show that what we solve is an *important open problem in a relevant regime*. For completeness, we also added these quotes to the Appendix A, that might useful to any other readers with the same doubts.
>
> - Bhandari et al. (2018), Section 8.1, on the complementarity of the robust and fast rate:
> > in the spirit of robust stochastic approximation [Nemirovski et al., 2009], the bound in part (a) gives a comparatively slow convergence rate of $\tilde{\mathcal{O}}(1/\sqrt{T})$, but where the bound and step-size sequence are independent of the conditioning of the feature covariance matrix $\Sigma$. The bound in part (c) gives a faster convergence rate in terms of the number of samples $T$, but the bound as well as the step-size sequence depend on the minimum eigenvalue $\omega$ of $\Sigma$.
>
> - Samsonov et al. (2024), Section 3, on the comparison with robust approaches:
> > **Comparison to the robust SA approach.** Note that the leading term of the bound in Theorem 3 includes factors of $1/\lambda_\text{min}$. This dependence is generally unavoidable if one aims to obtain the MSE bound for $\mathbb{E}[||\bar{\theta}\_n−\theta^\star||^2\_{\Sigma_\phi}]$ that scales as $1/n$. [...] In contrast, within the basin of robust stochastic approximation (RSA, (Nemirovski et al., 2009)), a convergence rate for $\mathbb{E}[||\bar{\theta}\_n−\theta^\star||^2\_{\Sigma\_\phi}]$ of order $O(1/\sqrt{n})$ can be derived with the instance-independent choice of step size. Importantly, this rate is not affected by a worst-case factor of $\lambda^{-1}_\text{min}$. This result was obtained for the TD algorithm in (Bhandari et al., 2018, Theorem 2).
>
> - Liu and Olshevsky (2021), Section 4.1, on the advantage of rates that are independent of the curvature:
> > One issue is the choice of step-size. The existing literature on temporal difference learning contains a range of possible step-sizes from $O(1/t)$ to $O(1/\sqrt{t})$ (see Bhandari et al. (2018); Dalal et al. (2018); Lakshminarayanan and Szepesvari (2018)). A step-size that scales as $O(1/\sqrt{t})$ is often preferred because, for faster decaying step-sizes, performance will scale with the smallest eigenvalue of $\Phi^\top D \Phi$ or related quantity, and these can be quite small. This is not the case, however, for a step-size that decays like $O(1/\sqrt{t})$.
>
> We hope this clarifies this serious misunderstanding and puts our results in the correct perspective.
>
> ---
>
> ### Answers to questions:
>
> **Monotonicity of rho_c**
>
> The proof that $\rho'_c\leq 0$ is tedious, but elementary. For completeness, we have now added it in Appendix Proposition F.3.
>
>
> **Possibility of almost-sure boundedness or high-probability bounds**
>
> Regarding the possibility of establishing almost-sure boundedness or high-probability bounds, we believe it is feasible (and we have promising ideas), but it requires a highly non-trivial Hoeffding-type inequality for Markov chains \[1\] to control the term
> $$\sum_{k=0}^{t-1} \eta_k \langle b_k, \theta_k - \theta^\star \rangle.$$
> As such, we leave such orthogonal direction to future work.
>
> **It is not clear how the self-bounding technique can be extended to TD($\lambda$) or Q-learning**
>
> When extending to TD($\lambda$) or Q-learning, one must verify the validity of Lemma 4.1, Lemma 4.5, and Lemma 4.6. These results indeed hold under the setting of Chen [2022, Prop. 3.1] (that we cite), which considers Q-learning under a sufficiently well-behaved behavior policy $\mu$, possibly with a relaxation of Lemma 4.1 to require only that
> $$
> \langle -\bar{g}(\theta), \theta - \theta^\star \rangle \ge 0.
> $$
>
> ---
> ### Unanswerable questions:
> We are unable to answer to the point about "Missing related references on finite-time analysis of TD(0)." because the reviewer does not list any reference.
> Once we receive concrete pointers, we will be happy to incorporate and discuss them accordingly.
>
>
> \[1\]  Hoeffding’s Inequality for General Markov Chains and Its Applications to Statistical Learning,  Jianqing Fan et al., JMLR 2021.

---

> > ### Comment · Reviewer_pgmn · 2025-11-26
> >
> > I thank the authors for their response and for addressing the points raised in the initial reviews. While I acknowledge the authors' efforts in correcting several identified technical issues, I beileve non-trivial issues in presentation and technical rigor likely remain. Further, although the general direction of relaxing strong convexity is indeed valuable, I am not yet convinced of the paper's novelty and technical impact relative to the established literature. The key algorithmic ideas and theoretical tools do not appear to be a significant advance over existing methods to warrant publication at ICLR. Finally, the provided bounds for Algorithm 1 depend on the initialization \theta_0 = 0. The authors' suggestion of using a doubling trick to mitigate the dependence on the time horizon T is therefore invalid, as this technique would inherently require re-initialization, violating assumption of the analysis.

---

> ### Author Response · Authors · 2025-11-26
>
> We thank the reviewer for their response. However, we found it extremely frustrating.
>
> **No additional or technical flaws are identified**, yet the reply states that "I believe ... likely remain" without providing *any* supporting evidence. Given that the reviewer did not find any technical flaws in our submission (only minor typos!), how is the judgment that "non-trivial issues likely remain" even warranted?
>
> We are also extremely confused by the sentence "The key algorithmic ideas and theoretical tools do not appear to be a significant advance over existing methods to warrant publication at ICLR." How can this be true given that **this was an open problem from 2018, and the key ideas of the proof are novel**?
>
> Regarding the doubling trick, it is fine to reset the algorithm with $\boldsymbol{0}$, this is exactly how the doubling trick is normally implemented. There is ample literature on this [1,2].
>
> Overall, **the only objection is the reviewer’s apparent dislike of the problem itself, despite seminal work explicitly identifying it as both important and open since 2018**. Given that this classic open problem was evidently unknown to the reviewer before reading our paper, and considering that such a problem is indeed important for this community, we respectfully argue that the current scores are not scientifically justified and should be reevaluated.
>
> ---
>
> [1] Online learning and online convex optimization, Shai Shalev-Shwartz, Foundations and Trends @ in Machine Learning, 2012
>
> [2] Online learning with predictable sequences, Alexander Rakhlin et al., COLT 2013

---

### Official Review · Reviewer_Fg3X · 2025-10-29

**Soundness:** 4
**Presentation:** 3
**Contribution:** 4
**Rating:** 10
**Confidence:** 5

**Summary:**

This paper investigates temporal-difference (TD) learning with linear function approximation without relying on projections or strong convexity. It removes a strong yet widely used assumption and, overall, makes a substantial contribution to the analysis of TD methods.

**Strengths:**

1. The authors establish non-asymptotic bounds for TD(0) with linear function approximation under Markov noise, matching practical projection-free  implementations.

2. The analysis shows the iterates remain bounded in expectation without projections or strong convexity, by leveraging a self-bounding structure in the recursion.

3. With a generic stepsize schedule, the method attains $ \tilde{O}\left(\frac{\\|\theta^\star\\|^2}{\sqrt{T}}\right)$,
  requiring no prior spectral/curvature information.

4. Using $f(\theta)= (1-\gamma)\left\lVert V_\theta - V_{\theta^\star}\right\rVert_{D}^{2}+ \gamma \left\lVert V_\theta - V_{\theta^\star}\right\rVert_{\mathrm{Dir}}^{2}$, which remains valid at $\gamma=1$ and aligns with the mean-path gradient, simplifies and strengthens the proof technique.

 5. The results remove reliance on projections/strong convexity and are supported by synthetic experiments that verify the existence of a stepsize threshold consistent with the theory.

**Weaknesses:**

The authors should provide intuition for their techniques—how they were derived and how they work in the proofs.

**Questions:**

1.  The authors should provide intuition for their techniques—how they were derived and how they work in the proofs.

2. Can your results be extended to adaptive settings (e.g., adaptive stepsizes) as in [1,2]?

3. Some references on  TD  [1,2] appear to be missing;.

[1] Adaptive temporal difference learning with linear function approximation.
[2] Finite-Time Analysis of Adaptive Temporal Difference Learning with Deep Neural Networks.

---

> ### Author Response · Authors · 2025-11-16
> **Thanks for your review**
>
> We thank the Reviewer for appreciating our work and perfectly summarizing our contributions. We hope that the Reviewer will help us in explaining the misunderstandings to the other Reviewers.
>
> ---
>
> ### Weaknesses:
>
> **Intuition of the proof**
>
> The intuition of our proof is simply summarized as follows:
> **If the iterate is already inside a sufficiently large ball centered at $\theta^\star$, then with a small enough stepsize the algorithm cannot drift outside this ball given the linear growth condition on the gradient.**
>
> Let's unpack this intuition.
>
> We want to show that with a suitably chosen stepsize, the linear TD algorithm enjoys *self-bounded iterates* **without relying on projection or strong convexity**. To motivate why this is feasible, let's start with **deterministic gradient** ascent on a convex function, with updates
> $$
> \theta_{t+1}=\theta_t+\eta_t g_t,
> $$
> where $\eta_t$ is the stepsize and $-g_t$ is the gradient of $f$ evaluated at $\theta_t$. Our objective is to show that $\lVert\theta_t\rVert_2$ remains bounded through all $t$, i.e., stays in a neighborhood of the maximizer $\theta^\star$.
>
> Let's proceed as follows:
> Using the triangle inequality, we have
> $$
> \lVert \theta_t \rVert_2 \le \lVert \theta_t-\theta^\star \rVert_2 + \lVert \theta^\star\rVert_2,
> $$
> so it suffices to control $\lVert\theta_t-\theta^\star\rVert_2$. From the basic expansion involving **linear growth condition on the gradient**
> $$
> \lVert \theta_t - \theta^\star \rVert_2^2 \leq \lVert \theta^\star \rVert_2^2 + \sum_{k=0}^{t-1} \eta_k^2 \lVert g_k \rVert_2^2 \leq \lVert\theta^\star \rVert_2^2 + c_g \sum_{k=0}^{t-1} \eta_k^2 \lVert \theta_k\rVert_2^2.$$
> Thus $\lVert\theta_t\rVert_2^2$ is controlled by
> $$
> \mathcal{O}\big(\max\\{ \lVert\theta^\star\rVert_2^2,\max_{k\le t-1}\lVert\theta_k\rVert_2^2 \\} \big),
> $$
> revealing a **self-bounding dependency**.
>
> This allows an inductive argument. Suppose inductively that
> $$
> \lVert\theta_k\rVert_2^2 \le \rho_c \lVert\theta^\star\rVert_2^2, \quad \forall k\leq t.
> $$
> We then show that, for sufficiently large $c$, the stepsize
> $$
> \eta_k=\frac{1}{c\log k\sqrt{k}}
> $$
> keeps the iterate within the same radius:
> $$
> \lVert\theta_{t+1}\rVert_2^2 \le \rho_c \lVert\theta^\star\rVert_2^2.
> $$
> Bounding the iterates thus reduces to solving an algebraic inequality involving $c$ and $\rho_c$.
>
> In the actual algorithm, gradients are not deterministic but **Markov-sampled**. This is handled using the standard coupling lemma (Bhandari et al. 2018, Lemma 11), which shows that the bias term also satisfies a bound of order
> $$
> \mathcal{O}\big(\max\\{ \lVert\theta^\star\rVert_2^2,\max_{k\le t-1}\lVert\theta_k\rVert_2^2 \\}\big).
> $$
>
>
> ---
> ### Answers to Questions
>
> **Missing references and potential extensions**
>
> Thanks for pointing out these interesting papers, we have now added them to the Related Work section and one of them to the Table 1.
> In the following, we comment on the possibility of extending our approach to their algorithms.
>
> [1] **Adaptive Temporal Difference Learning with Linear Function Approximation**
>
> In this work, the Projected Adaptive TD(0) algorithm uses an Adam-style update of the form
> $$
> \theta^{k+1}
>   = \mathrm{Proj}_R\left(\theta^k + \eta \frac{m^k}{\sqrt{v^k+\delta}}\right).
> $$
> We believe our proof technique can be adapted to this setting to eliminate the projection step, provided the stepsize $\eta$ is chosen carefully.
>
> Indeed, observe that the term $m^k / \sqrt{v^k + \delta}$ has magnitude on the order of $O(k^{-1/2})$.
> Our analysis fundamentally relies on the fact that in linear TD, the stochastic gradient satisfies $g_t = O(\lVert\theta_t\rVert_2)$.
> Therefore, a natural choice in the adaptive setting is to let the effective stepsize scale as
> $$
> \eta_t = \max_{k \le t}\lVert\theta_k\rVert_2,
> $$
> which would ensure compatibility with the self-bounding structure used in our proof.
>
>
> [2] **Finite-Time Analysis of Adaptive Temporal Difference Learning with Deep Neural Networks**
> This work extends the projected adaptive TD(0) method from linear function approximation to neural network approximation via NTK theory. We are not sure how our proof would interact with the NTK parametrization, so applying our proof strategy in this setting may introduce nontrivial challenges. Nonetheless, replacing the constant stepsize with a time-dependent choice such as
> $$
> \eta_t = \max_{k\le t}\lVert\theta_k\rVert_2
> $$
> appears to be a plausible starting point for removing the projection in the Neural TD framework.

---

> > ### Comment · Reviewer_Fg3X · 2025-11-17
> > **reply**
> >
> > Thanks for your response. I will keep my score as is. I believe this paper is suitable for publication at ICLR.

---

### Official Review · Reviewer_6z2S · 2025-11-01

**Soundness:** 3
**Presentation:** 3
**Contribution:** 3
**Rating:** 0
**Confidence:** 4

**Summary:**

Projection-free finite-time analysis of TD(0) with linear features under Markovian sampling. Uses a Liu–Olshevsky–style potential and a horizon-aware stepsize to prove self-bounded iterates and a robust $\tilde{\mathcal{O}}(1/\sqrt{T})$ rate; small synthetic experiments illustrate a stepsize threshold.

**Strengths:**

- First robust, **projection-free** guarantee for linear TD(0).
- Handles Markovian bias via mixing; clear comparison to prior work.

**Weaknesses:**

- **The paper clearly violates the ICLR format. Missing page numbers and the whole text has been shifted down and therefore should be desk rejected.**
- Experiments are narrow; few task details in main text.

**Questions:**

None

---

> ### Author Response · Authors · 2025-11-16
> **Latex package conflicted, fixed now**
>
> We thank the reviewer for pointing out this issue: We now realized that we used a latex package that interfered with the ICLR style. Overall, this made us gain ~10 lines of text without us noticing it. We apologize for this stupid mistake.
>
> Luckily, ICLR allows changes to the submission, so we have already fixed the pdf on OpenReview, by changing some display equations to inline to remove the additional 10 lines.
>
> That said, the ICLR Reviewer's guidelines (https://iclr.cc/Conferences/2025/ReviewerGuide) say
>
> >Q: How should I handle a policy violation ?
>
> >A: To flag a CoE violation related to a submission, please indicate it when submitting the CoE report for that paper. The AC will work with the PC and the ethics board to resolve the case. To discuss other violations (e.g. plagiarism, double submission, paper length, formatting, etc.), please contact either the AC/SAC or the PC as appropriate. You can do this by sending a confidential comment with the appropriate readership restrictions.
>
> Hence, if the AC/SAC or PC decide that this violation not critical, given that we fixed this (very minor) issue in the updated pdf, we invite the reviewer to update their review as well.
> We already contacted the AC, SAC, and PC to be sure the correct protocol is followed and we are either desk-rejected or we receive an updated review. We are confident that the reviewer will adhere to the guidelines.

---

### Meta-Review · Area_Chair_HpVf · 2026-01-07

**Summary:**

In this paper, the authors study finite-time convergence of TD(0) with linear function approximation under Markovian sampling, without the use of projection to enforce bounded iterates. Building on a Liu-Olshevsky-style potential that combines a stationary-weighted squared error and a Dirichlet-type semi-norm, they propose a horizon-aware diminishing step-size and prove a robust $\tilde{O}(1/\sqrt{T})$ convergence guarantee in the presence of Markovian noise. A key technical contribution is a self-bounding argument that controls the iterates in expectation (without explicit projections). They also include a synthetic experiment to illustrate a step-size-threshold phenomenon.

- The paper targets an important gap between theory and practice: TD(0) is typically run without projection, while many robust guarantees rely on projection.

- The authors use an appropriate potential (stationary $D$-norm error plus a Dirichlet seminorm) that yields clean identities for mean TD dynamics and is well-matched to robust analysis under Markovian sampling.

- They provide a clear comparison to prior work, distinguishing robust guarantees from curvature-dependent fast rates and explaining where projection enters earlier analyses.

- The proof of iterate boundedness relies on an inductive step where the next iterate is bounded by the same constant as previous iterates. This ``tangled recursion'' is resolved by choosing a very small step-size. This may lead to extremely slow progress in the early stages of learning, a trade-off for stability that is not quantitatively compared to projected methods in the main text.

- $\xi_{t-1}$ and $b_{t-1}$ are introduced as a martingale difference and the bias. The reliance on Lemma 4.5 to control this bias requires TD(0) to effectively wait for the Markov chain to mix. In environments with slow mixing, the $\tilde{O}(1/\sqrt{T})$ rate might be dominated by the mixing time constants for a very long period. This means a dependency on ''curvature of the objective'' is swapped with a dependency on ''mixing of the chain'', both unknown quantities in most problems.

- The authors argue in Section 5 that knowing $T$ in advance is not a limitation because of the doubling trick. However, the self-bounding induction (Theorem 4.2a) is specifically calibrated to the $\log T$ term in the denominator of $\eta_t$. If $T$ is unknown and a doubling trick is used, the step-size would change discontinuously at each doubling epoch. The paper does not prove that the ''self-bounding'' property holds across these epoch transitions, where the previous iterates were generated with a different $T$ assumption. I believe an explicit proof/discussion would make this claim stronger.

- A claim in the bias discussion does not seem to be written correctly:

$$ \mathbb{E}_{Z_k \sim (\pi , \pi P^\mu)} [ g(\theta_k , Z_k)] \neq \bar{g}(\theta_k) $$.

The intended point appears to be about conditional expectations along the trajectory and the dependence between $\theta_k$ and the sampled transition, but the current statement mixes stationarity and trajectory dependence in a way that seems imprecise. Rewriting this portion with explicit conditioning would substantially improve clarity.

- It would be good to rewrite the bias discussion using explicit conditioning and clearly distinguishing a fresh stationary draw from a trajectory sample, so the mathematical statement matches the intended argument.

- It would be good to have a discussion on how sensitive the bounds and the observed step-size threshold are to the choice of $\tau = \tau(1/\sqrt{T})$. To clarify if this is primarily an analysis convenience or reflects the empirically relevant regime.

I believe the paper has novel contributions but it has been written in a way that raises many questions/concerns for readers/reviewers, especially those that are not hardcore (RL) theoreticians. Improving the formulation, having more discussion to clarify certain parts of the analysis/assumptions/claims, and most importantly discussing/highlighting the limitations of the results (in comparison to the existing ones) would significantly improve the work and make it more accessible to a wider audience.

**Reviewer Concerns:**

The authors' response did not address the concerns by Reviewer pgmn.

**Reviewer Scores:**

Reviewers Fg3X gave the paper the maximum score of 10 with maximum confidence of 5 and maintained their score after the authors' response.

Reviewer t624 gave score 2 with confidence 5 and did not participate in the discussion. It is hard to say if the reviewer would have changed their score.

Reviewer pgmn gave score 2 with confidence 4 and was not convinced by the authors' response and maintained their score.

Reviewer 6z2S who had initially gave score 0 due to a format violation without consulting with the ACs/SACs/PCs, reviewed the paper after the rebuttals and gave it score 6 with confidence 4.

---

### Decision · Program_Chairs · 2026-01-26

Reject